# The Fate of Sialic Acid and PEG Modified Epirubicin Liposomes in Aged versus Young Cells and Tumor Mice Models

**DOI:** 10.3390/pharmaceutics14030545

**Published:** 2022-02-28

**Authors:** Dezhi Sui, Xianmin Meng, Changzhi Li, Xueying Tang, Ying Qin, Ning Zhang, Junqiang Ding, Xinrong Liu, Yihui Deng, Yanzhi Song

**Affiliations:** College of Pharmacy, Shenyang Pharmaceutical University, Wenhua Road, No. 103, Shenyang 110016, China; suidezhi741@163.com (D.S.); MXM20220129@163.com (X.M.); lichangzhi365@163.com (C.L.); txy6668882022@163.com (X.T.); 15841408894@163.com (Y.Q.); zncsspu@163.com (N.Z.); junqiang_ding@163.com (J.D.); yaojixueyaoxue@163.com (X.L.); dengyihui@syphu.edu.cn (Y.D.)

**Keywords:** age, epirubicin, sialic acid, PEG, PBMs, liposomes

## Abstract

In preclinical studies of young mice, nanoparticles showed excellent anti-tumor therapeutic effects by harnessing Peripheral Blood Monocytes (PBMs) and evading the immune system. However, the changes of age will inevitably affect PBMs and the immune system, and there is a serious lack of relevant research. Sialic acid (SA)-octadecylamine (ODA) was synthesized, and SA- or polyethylene glycol (PEG)-modified epirubicin (EPI) liposomes (EPI-SL and EPI-PL, respectively) were prepared to explore differences in antitumor treatment using 8-month-old and 8-week-old Kunming mice. Based on presented data, 8-month-old mice had more PBMs in peripheral blood than 8-week-old mice, and age differences resulted in different anti-tumor treatment effects following EPI-SL and EPI-PL treatment. Following EPI-PL administration, the tumor volume was significantly smaller in 8-week-old mice than in 8-month-old mice (* *p* < 0.05). Eight-month-old mice treated with EPI-SL (8M-SL) presented no damage to healthy tissue, with a 100% survival rate, and 50% mice in 8M-SL showed ‘shedding’ of tumor tissues from the growth site. Accordingly, 8-month-old mice treated with EPI-SL achieved the best therapeutic effect at different ages and with different liposomes. EPI-SL could improve the antitumor effect of 8-week-old and 8-month-old mice.

## 1. Introduction

Older individuals, particularly those that are frail or those living in long-term care facilities, are disproportionately affected by diseases (especially cancer), and the aging of tissues in the body has serious effects on therapy [1,2,3] as aging leads to a series of degenerative changes in the function and number of innate immune cells (monocytes, macrophages, etc.) [4]. Cytotoxicity, phagocytosis, and antigen presentations of the cells were significantly changed. Monocytes are the largest white blood cells in the blood and are an important part of the body’s defense system [5]. Monocytes participate in the immune response. After phagocytosing antigens, they carry the antigenic determinants to lymphocytes, induce specific immune responses of lymphocytes, and have the ability to recognize and kill tumor cells [6,7]. After the changes of age, the ability of monocytes to migrate back to the lesion site; phagocytosis of pathogens; free radical production; and intracellular killing function decreased [8,9]. Changes of PBMs will inevitably affect the distribution and therapeutic effects of nanoparticles in the body [10].

Pre-clinical anti-tumor studies all use 4–8 weeks old mice as experimental animal models, which seriously ignores the clinical age distribution of patients [11,12]. Over the last two to three decades, several nanoparticles have been designed based on immune system evasion, with polyethylene glycol (PEG)-modified nanoparticles emerging as the primary type. Doxil^®^ was the first marketed liposomal product using the enhanced permeability and retention (EPR) effect of tumor sites [13,14]. For this formulation, the particle size is controlled below 100 nm, while PEG on the surface of Doxil^®^ forms a hydration layer to avoid uptake by immune cells, thereby increasing the residence time in peripheral blood [15]. Furthermore, its passage across the incomplete structure of the tumor-related blood vessel wall enables passive accumulation in the tumor to inhibit tumor growth [16]. In recent years, the PBMs drug delivery system has been widely studied and used due to its superior therapeutic efficacy. A drug delivery strategy of harnessing the immune system has emerged, affecting tumor immune circulation by targeting immune cells [17]. Our laboratory has long focused on the study of the interaction between immune cells and nanoparticles. Additionally, SA derivatives (SA-ODA) have been synthesized as targets of SA-modified nanoparticles that specifically achieve the PBMs drug delivery system [18]. SA-modified liposomes are reportedly recognized by the Siglec-1 receptor on the monocyte surface, phagocytosed, and delivered to the tumor site through spontaneous migration [19]. This strategy increases drug accumulation in the tumor, destroys the tumor immunosuppressive state, and enhances infiltration of PBMs (some mice even shed tumors). SA-modified liposomes and Doxil^®^ are delivered to the tumor by “harnessing” and “evading” immune cells, respectively. However, the results of the above experimental studies are based on mice aged 4–8 weeks. Whether the therapeutic effect of SA-modified nanoparticles and PEG-modified nanoparticles is altered by age remains unclear [20].

Currently, there is a lack of research assessing differences in efficacy between SA-modified epirubicin liposomes (EPI-SL) and PEG-modified epirubicin liposomes (EPI-PL), for patients with cancer in different age groups. Significant differences in tumor molecular characteristics between patient populations of different ages can lead to a decrease in the anti-therapeutic effect of nanoparticles. To assess the influence of age on different anti-tumor mechanisms (e.g., harness and evade), PEG-modified epirubicin liposomes (EPI-PL) and EPI-SL were prepared using an (NH_4_)_2_SO_4_ gradient [21]. S180 cells were injected into the armpit of 8-week-old and 8-month-old Kunming mice to establish tumor models [22,23]. Based on presented data, 8-month-old Kunming mice exhibited superior therapeutic efficacy, less hair loss, and minimal tissue damage following EPI-SL treatment when compared with mice administered EPI-PL.

## 2. Materials, Animals, and Methods

### 2.1. Materials and Animals

EPI was purchased from Olympic Star Pharmaceutical Co., Ltd. (Shenzhen, China). SA was obtained from Changxing Pharmaceutical Co., Ltd. (Huzhou, China). Hydrogenated soy phosphatidylcholine (HSPC) and 1,2-distearoyl-sn-glycero-3-phosphoethanolamine-*N*-[amino(polyethylene glycol)-2000] (mPEG2000-DSPE) were procured from A.V.T. Pharmaceutical Co., Ltd. (Shanghai, China). ODA and 1,1′-dioctadecyl-3,3,3′,3′-tetramethylindotricarbocyanine iodide (DiR) were obtained from Molecular Probes, Inc. (Eugene, OR, USA). *N*-hydroxysuccinimide (NHS), cholesterol (CH), *N*-(3-dimethylaminopropyl)-*N*-ethylcarbodiimide HCl (EDC·HCl), and triethylamine (TEA) were purchased from China National Medicines Co. Ltd. (Shenyang, China). SA were purchased from Shanghai Macleans Biochemical Technology Co., Ltd. (Shanghai, China). Anti-mouse CD169-APC, anti-mouse CD115-FITC, and anti-mouse CD8-PE were obtained from eBiosciences (San Diego, CA, USA). Other chemical reagents were of high-pressure liquid chromatography (HPLC) or analytical grade.

RAW264.7 mouse macrophages and S180 sarcoma cells were purchased from the Chinese Academy of Sciences (Shanghai, China). The cell culture medium was composed of RPMI 1640 medium and Gemini Foundation fetal bovine serum (FBS) (9:1, *v*/*v*) (Meilun Co., Ltd. Dalian, China).

Male Kunming mice (8-month-old and 8-week-old) were procured from the Experimental Animal Center of Shenyang Pharmaceutical University (Shenyang, China). All procedures performed during the experimental period complied with guidelines and laws regarding animal feed and use, as formulated by Shenyang Pharmaceutical University Animal Studies Committee (SYPU-IACUC-C2020-12-3-208, 3/12/2020). The left underarms of Kunming mice (8-week-old or 8-month-old) were inoculated with 1 × 10^7^ suspended S180 cells.

### 2.2. Peripheral Blood Monocytes in Aged and Young Mice

#### 2.2.1. Isolation of Peripheral Blood Monocytes

PBMs were separated from peripheral blood collected from aged and young mice using a test kit (Tianjin Hao Yang Biological Manufacture Co., Ltd., Tianjin, China). In brief, peripheral blood samples of different-aged mice were collected; then, erythrocyte sedimentation fluid was mixed with blood samples (1:2, *v*/*v*). The supernatant consisting of PBMs was obtained after 30 min. The cell suspension was dropped onto the surface of a mixture medium (TBD2011M, Tianjin Hao Yang Biological Manufacture Co., Ltd., Tianjin, China), including PBM separation solutions 1 and 2 in the test kit (3/1, *v*/*v*). PBMs were trapped between the liquid interface of the diluent and separation solution 2 after centrifugation at 800 g for 25 min. Then, PBMs were characterized using the allophycocyanin (FITC)-conjugated CD115 antibody. The total number of isolated PBMs in the blood was counted with a hemocytometer.

#### 2.2.2. Volume and Density of Peripheral Circulating Blood

Blood samples (1 mL) were harvested by performing a retroorbital sinus puncture, and the weight of 1 mL of the blood sample was measured. One milliliter of blood collected from 8-month-old and 8-week-old mice was used to calculate blood density as the ratio of weight to volume. Blood collected from the abdominal aorta was used to obtain blood from peripheral circulation for different-aged mice. The weight of the blood was converted into a volume value using the corresponding density.

### 2.3. Chemical Synthesis of SA-ODA

SA-ODA was synthesized by reacting the carboxyl group of SA with the amino group of ODA. SA (6.9 g) was added to 100 mL of DMF at 70 °C and stirred for 10 min to dissolve the SA. Then, 5.25 g of NHS, 6.3 mL of TEA, and 8.55 g of EDC·HCl were added to the SA-solution, cooled to 25 °C, and stirred at 0 °C for 1 h. Next, 2.1 g of ODA was added to the SA-solution, which was heated to 60 °C and reacted for 12 h at 60 °C to synthesize the crude product. Finally, a 1 kDa dialysis bag was used to remove impurities from the crude product to obtain the final SA-ODA product. SA-ODA was verified by 1H-nuclear magnetic resonance (NMR; Bruker 600-MHz, Bruker Corporation, Billerica, MA, USA) and mass spectroscopy (MS; Agilent 1100, Bruker Corporation, Billerica, MA, USA).

### 2.4. Molecular Docking of Siglec-1 Receptor and SA-ODA

AutoDock Vina was employed to calculate the affinity energy of the interaction between the Siglec-1 receptor and SA-ODA (or SA). The Protein Data Bank (http://www.rcsb.org/, accessed on 10 March 2021) was used to download 1URL, and Siglec-1 was separated from 1URL using PyMOL 2.0. PyMOL 2.0 was used to select Arg105, Leu107, and Arg97 for cavity positioning. ChemDraw 19.0 was used to draw SA-ODA, and AutoDock Vina was used for docking the selected 1URL cavity. The final data are presented using PyMOL 2.0 and ZBH—Center for Bioinformatics (https://proteins.plus/, accessed on 12 March 2021) [24].

### 2.5. Preparation of EPI-SL and EPI-PL

The thermal ethanol injection method was used to prepare liposomes. HSPC, CH, and other components (SA-ODA or mPEG_2000_-DSPE) were dissolved in absolute ethanol, which was heated to 60 °C and mixed uniformly. Ethanol that dissolved the membrane material was evaporated to form a transparent film; then, an (NH_4_)_2_SO_4_ solution (200 mmol/mL) was added and stirred for 30 min. The hydrated phospholipid was sonicated at 200 W for 2 min and 400 W for 6 min using an ultrasonic cell pulverizer (JY92-IIDN; Ningbo Scientz Biotechnology Co., Ltd., Ningbo, China). For sterilization, a 0.22 µm microporous membrane was employed. Next, 100 μL of liposomes was dropped to the surface of Sephadex^®^ G-50 column, allowed to stand for 1 min, centrifuged (1600 g) for 4 min, and washed twice with distilled water (50 μL), with centrifugation (1600 g) for 4 min to obtain blank liposomes. Blank liposomes in the aqueous phase without ammonium sulfate were incubated with the EPI solutions (5 mg/mL) and stirred for 20 min. A Sephadex^®^ G-50 column was used to remove free EPI. Then, DiR was dissolved in absolute ethanol (10 mg/mL) and incubated with 2 mL EPI-SL or EPI-PL for 20 min to prepare SA-modified DiR-labeled EPI liposomes (DE-SL) and PEG-modified DiR-labeled EPI liposomes (DE-SL), respectively.

### 2.6. Characterization of EPI-SL and EPI-PL

A NICOMP 380 HPL submicron particle analyzer (Particle Sizing Systems, Santa Barbara, CA, USA) was used to determine zeta potential and particle size of EPI-SL, EPI-PL, DE-SL, and DE-PL. Transmission electron microscopy (TEM; JEM-2100, JEOL Co., Ltd., Tokyo, Japan) was used to assess the morphology of EPI-PL and EPI-SL. Accordingly, EPI-SL and EPI-PL were diluted 200-fold with sterilized water for injection; diluted solutions were added to the copper net (200 mesh) and then negatively stained (after 5 min) with 2% phosphotungstic acid dye solution for 4 min. Finally, samples were subjected to TEM.

The encapsulation efficiency (EE%) of liposomes was determined using the molecular sieve method. In brief, 100 μL EPI-SL or EPI-PL was placed on a Sephadex^®^ G-50 column and washed twice with 50 μL distilled water to obtain 200 μL EPI-SL or EPI-PL without the free drug. After the column process, EPI-SL or EPI-PL (100 μL) and liposomes (200 μL) were separately placed in a 10 mL volumetric flask, along with 90% isopropyl alcohol containing 1.0 M HCl for pipetting and demulsifying the solution. A UV-1801 UV/VIS spectrophotometer (Beijing Rayleigh Analytical Instrument Co., Ltd., Beijing, China) was used to measure absorbance at 480 nm. The EE% was calculated using the following formula:EE% = The ratio of after column (A_a_) to before column (A_b_) (A_a_/A_b_) × 100%

Long-term stability tests were conducted at 4 ± 2 °C in accordance with drug stability testing guidelines and with reference to the storage conditions of the marketed liposome product, Doxil^®^. Three batches of EPI-SL, EPI-PL, DE-SL, and DE-PL were prepared, sealed with nitrogen, and stored at 4 ± 2 °C in the dark. Samples were collected at 0, 1, 2, and 3 months, and the changes in particle size and encapsulation rate of EPI-SL, EPI-PL, DE-SL, and DE-PL were recorded.

### 2.7. Drug Release of EPI-SL, EPI-PL, and EPI-Solution

The isotonic pressure dialysis method was used to calculate the cumulative drug release rate. EPI-SL, EPI-PL, and EPI-solution (EPI-S, 4 mL, 1 mg/mL) were separately placed in 100 mL of release media (0.2 M histidine and 2 M ammonium chloride; pH = 6.5) using 5 kDa dialysis bags and stirred at constant speed at 37 °C. Finally, 4 mL of the sample was withdrawn, 4 mL of fresh release media was added, and absorbance was measured at 480 nm. The cumulative release rate (R_n_) was calculated using the following formula:$$R_{n}=(C_{n}V_{o}+\sum_{n-1}^{n}C_{n-1}V)/M_{t}\times 100\%$$
where M_t_ is the total mass of the drug in the dialysis bag, V_o_ = 100 mL, and V = 4 mL; concentration of the drug was calculated by scaling (Cn).

### 2.8. Cytotoxicity of EPI-SL, EPI-PL, and EPI-Solution

In brief, 100 μL suspended S180 or RAW264.7 cells (10^5^ cells/well) were seeded on a 96-well plate and cultivated for 12 h (37 °C, 5% CO_2_). Thereafter, 10 μL of 0, 1, 10, 20, 50, and 100 μg/mL EPI-SL, EPI-PL, or EPI-solution (EPI-S) was added to the cells, followed by 50 μg/well of MTT after 24 h. The optical density (OD) was recorded at 570 nm using a microplate reader (Model 680, Bio-Rad, Hercules, CA, USA).

Inhibition rate (IR) = 1 − OD value difference between the preparation and zero group/OD value differences between control and blank groups.

### 2.9. Uptake Ability of PBMs and RAW264.7 Cells

RAW264.7 cells were seeded on a 6-well plate and cultured in a 5% CO_2_ incubator at 37 °C for 2 h. Next, cells were cultivated with EPI-SL or EPI-PL (final concentration of EPI: 15 μg/mL). After 2 h of incubation, the cells were fixed with 4% paraformaldehyde. DAPI (50 μg/mL) was added and incubated for 20 min in the dark. RAW264.7 cells were cultivated with SA (SA_pre_) solution for 12 h, repeating the above experiment.

Isolated PBMs in aged and young mice were seeded on a 6-well plate and cultured in a 5% CO_2_ incubator at 37 °C for 2 h. Next, cells were cultivated with EPI-SL and EPI-PL (final concentration of EPI: 15 μg/mL). After 2 h of incubation, the cells were fixed with 4% paraformaldehyde. DAPI (50 μg/mL) and allophycocyanin (APC)-conjugated CD 169 antibody (Biolegend, San Diego, CA, USA) were subsequently added and incubated for 20 min in the dark. A confocal laser scanning microscope (Carl Zeiss, Jena, Germany) was used to observe cells. A flow cytometer (Beckman Coulter, Fullerton, CA, USA) was used to detect and analyze the fluorescence intensity EPI in cells.

### 2.10. Pharmacokinetics of EPI-SL and EPI-PL in Aged and Young Mice

Accordingly, 48 tumor-bearing mice (8-month-old) were randomly divided into two treatment groups (n = 12): EPI-SL injected group (8M-SL) and EPI-PL injected group (8M-PL). In addition, 48 tumor-bearing, 8-week-old mice were randomly divided into two treatment groups (n = 12): EPI-SL-injected group (8W-SL) and EPI-PL injected group (8W-PL). The EPI dose was 5 mg/kg. Blood samples (1 mL) were collected by retroorbital sinus puncture from each group at 0.0167, 0.5, 1, 2, 4, 8, 12, and 24 h post-treatment (n = 3). The collected blood sample was placed in a heparin anticoagulation tube to obtain fresh anticoagulated blood; 100 μL of RIPA lysate, and mix well, were added; 800 μL of ethanol was added, vortexed for 5 min, and centrifuged at 10,000× *g* for 10 min; 200 μL of the supernatant was processed and added to a 96-well plate; and a microplate reader (Model 680, Bio-Rad, USA) was used to measure the OD value at λ_ex_ = 482 nm and λ_em_ = 590 nm. DAS 2.1.1 pharmacokinetic software was used to analyze the pharmacokinetic parameters.

Repeating the above experiment, blood samples (1 mL) were collected by retroorbital sinus puncture from each group at 2 h post-treatment (n = 3). PBMs were separated from peripheral blood collected from aged and young mice using a test kit, and a flow cytometer was used to detect and analyze the fluorescence intensity of EPI and anti-mouse CD115-FITC in cells.

### 2.11. Tissue Distribution of DE-SL and DE-PL in Aged and Young Mice

We investigated the effects of SA, PEG, and age on liposome distribution in vivo. Accordingly, 12 tumor-bearing mice (8-month-old) were randomly selected and divided into two groups: DE-SL injected group (8M-SL) and DE-PL injected group (8M-PL). Additionally, other 12 tumor-bearing, 8-week-old mice were randomly divided into two groups: the DE-SL injected group (8W-SL) and the DE-PL injected group (8W-PL). Mice were administered DE-SL and DE-PL through the caudal vein (0.2 mg/kg). From each group, three mice were sacrificed at 2, 8, 12, and 24 h post-treatment, and FX Pro In Vivo Imaging System (Carestream Molecular Imaging, Woodbridge, CT, USA) was used to obtain fluorescence images. The tumor, heart, liver, spleen, kidneys, and lung were harvested for fluorescence imaging and quantitative analysis of DiR-labeled liposomes in the main organs of 8-month-old and 8-week-old mice.

### 2.12. Pharmacodynamics of EPI-SL and EPI-PL in Aged and Young Mice

The left underarm of 18 Kunming mice (8-week-old) was inoculated with 1 × 10^7^ suspended S180 cells. A total of three groups were established: control (8W-Con), EPI-SL injection (8W-SL), and EPI-PL injection (8W-PL); each group was composed of six mice. In addition, the same treatment protocol and groups were modeled in 8-month-old mice, inoculated with 1 × 10^7^ suspended S180 cells into the left underarm (n = 6): control (8M-Con), EPI-SL injection (8M-SL), and EPI-PL injection (8M-PL). Each dose was injected every three days for a total of five doses (EPI, 5 mg/kg). During treatment, long (a) and short (b) diameters of the tumor, body weight, and tumor volume (V_t_ = 0.5 × a × b^2^) were measured daily.

In order to investigate the organ damage induced by off-target liposomes in aged and young mice, pathological sections of the main organs were prepared. In brief, 4% paraformaldehyde was used to fix the spleen, livers, thymus, heart, tumors, kidneys, and lungs of mice; then, the samples were embedded in paraffin blocks. An ultrathin microtome was used to cut 5-μm sections of embedded specimens, with paraffin removed from sections at 70 °C. Hematoxylin-eosin (H&E), anti-mouse CD8-PE, and TUNEL reagent (Roche, Basel, Switzerland) were used to stain the samples, and an inverted microscope was used to observe the samples.

### 2.13. Statistical Analysis

A one-way ANOVA followed by a post hoc test was used for multiple group comparisons, and Student’s t-test (two-tailed) was used for comparisons of two groups by using SPSS 23.0 (IBM Corp., Armonk, NY, USA). GraphPad Prism 6 (GraphPad Software, Inc., San Diego, CA, USA) was used to draw figures and tables. The fluorescence intensity was calculated using ImageJ (National Institutes of Health, Bethesda, MD, USA). Data values are expressed as mean ± standard deviation. * *p* ≤ 0.05, ** *p* ≤ 0.01, and *** *p* ≤ 0.001 were considered significant.

## 3. Results and Discussion

### 3.1. Characterization and Molecular Docking of SA-ODA

MS and 1H NMR were used to analyze SA-ODA. Based on MS findings, 595.4 Da [SA-ODA+Cl]^−^ and 583.4 Da [SA-ODA+Na]^+^ were mainly detected. According to 1H NMR (CD3OD, dppm), 3.93 (2H, H-6, H-4), 3.71 (2H, H-9, H-5), 3.58 (2H, H-90, H-8), 3.21 (1H, H-7), 3.10 (2H, H-1), 2.19 (1H, H-30), 1.92 (3H, H-10), 1.43 (1H, H-3), 1.19 (30H, alkyl), and 0.8 (3H, H-11) were identified (Figure 1A). These results confirmed the successful synthesis of SA-ODA. Hydrogen bonding combined SA-ODA and H of the ARG105 and ARG97 residues in the Siglec-1 receptor (Figure 1B). The affinity energy of SA-ODA was −4.6 kcal/mol, and the energy was <0 kcal/mol, so SA-ODA could combine with the Siglec-1 receptor.

SA was highly hydrophilic, and lipid derivatization was required to anchor SA on the liposomal surface [25,26]. In 2017, Zhou et al., coupled the carboxyl group of CH and the hydroxyl group of SA to form an SA derivative, which significantly increased the uptake of tumor-associated macrophages [27,28]. This experiment was designed to synthesize SA-ODA with a molecular weight of 560 kDa by coupling the amino group (NH_2_) of ODA with the carboxyl group (HOOC) of SA. SA-ODA could be purified to a concentration of 99% by washing and recrystallization and could specifically recognize and bind to the Siglec-1 receptors of PBMs.

### 3.2. Characterization of EPI-SL and EPI-PL

Physical characteristics of liposomes, including EE%, morphology, cumulative release rate, particle size, and zeta potential, are crucial parameters for evaluating the effectiveness of the liposomal formulation in vivo (Table 1). The phospholipid composition of the liposomes is shown in Table 1. The zeta potential was −15 to −35 mV, and the EE% of EPI was greater than 95%. The particle size of all liposomes ranged between 105 and 115 nm, and the particle sizes based on the images captured using a transmission electron microscope were essentially identical. Furthermore, EE% and particle size showed no significant difference after three months (Table 1 and Figure 2A,B). However, low-solubility aggregates were generated in the internal water phase of EPI-SL and EPI-PL, demonstrating coffee bean particles (red arrow, Figure 2C). The release rates of the PEG and SA modes were significantly slower than that of the EPI-solution. The cumulative release of EPI-SL and EPI-PL within 48 h was less than 20% (Figure 2D).

The thermal ethanol injection method eliminated the toxicity of residual organic reagents. For the first marketed liposome, i.e., Doxil^®^, the ammonium sulfate gradient method was used to load EPI. A low-solubility polymer was formed by integrating sulfate radicals with the amino groups for EPI; the EE% of EPI was above 95%. The presence of the liposome phospholipid bilayer effectively delayed the EPI release rate and improved the stability of liposome, and there was no change in liposome within three months.

### 3.3. Uptake and Cytotoxicity In Vitro

For RAW264.7 and S180 cells, the IC_50_ value of EPI-SL was 5–7 times lower than that of EPI-PL (Table 2). Sialic acid on the surface of liposomes could increase the uptake of liposomes by cells with Siglec-1 receptors, and PEG on the surface of liposomes could reduce the uptake of liposomes by cells (Figure 3A–C). The cell-associated fluorescence intensity of EPI in EPI-SL was significantly higher than those of EPI in EPI-PL and EPI-SL(SA) (*** *p* ≤ 0.001). There was no significant difference between the cell associated fluorescence intensity of EPI in EPI-PL and EPI-PL(SA). PBMs were obtained from peripheral blood samples using the density gradient centrifugation method. The harvest was close to 6 × 10^5^ cells/mouse for 8-week-old mice and 1 × 10^6^ cells/mouse for 8-month-old mice; more than 90% of the PBMs were active (Figure 3D–F).

For PBMs of aged and young mice, the expression of the Siglec-1 receptor of isolated PBMs was significantly higher in 8-month-old mice than in 8-week-old mice, and the fluorescence intensity of EPI in EPI-SL and EPI-PL was significantly higher in 8-month-old mice than in 8-week-old mice (* *p* ≤ 0.001, *** *p* ≤ 0.001, Figure 4A–D). The results of the flow cytometer measurement were consistent with the results of the confocal microscope measurement, indicating that the expression of Siglec-1 receptor on the surface of single cells increased and the uptake capacity of single cells was enhanced in 8-month-old mice.

PEG formed space steric hindrance on the surface of EPI-PL, causing steric hindrance that reduced cell uptake and cytotoxicity [29]. SA binds explicitly to the Siglec-1 receptor on the cell surface, increasing the uptake of EPI-SL by RAW264.7 and PBMs with high Siglec-1 receptor expression [30]. The IC50 value of EPI-PL was higher than that of EPI-SL. With the increase of age, the expression of Siglec-1 receptor on the surface of PBMs of 8-month-old mice increased. For SA-modified liposomes, the higher the expression of Siglec-1 receptor, the greater the uptake of a single cell. For PEG modified liposomes, the stronger the phagocytic ability of the cell, the greater the uptake of a single cell [31].

### 3.4. Tissue Distribution of EPI-SL and EPI-PL in Aged and Young Mice

With the change of age, the total volume of the peripheral blood of the mouse and the number of immune cells in the peripheral blood will change. For 8-month-old or 8-week-old mice, the concentration of EPI in plasma of mice injected with EPI-PL at the same time point was significantly higher than that of mice injected with EPI-SL within 24 h (Figure 5A,B). For EPI-SL and EPI-PL, fluorescence intensity of EPI was significantly higher in the PBMs of 8-week-old mice at 2 h than of 8-month-old mice (Figure 5C). The fluorescence intensity of DiR in tumor and the liver at 2 and 8 h of mice injected with DE-SL was significantly higher than that of DE-PL, irrespective of mice age. For DE-SL, DiR accumulation was significantly higher in livers of 8-week-old mice at 2 and 8 h than of 8-month-old mice. In tumors, fluorescence intensity of DiR of 8-month-old mice was significantly higher than that of 8-week-old mice at 2 and 12 h, and DiR accumulation attained a maximum value in 8M-SL at 8 h (Figure 5D,E). For DE-PL, fluorescence intensity of DiR was significantly higher in the tumor of 8-week-old mice at 12 h than of 8-month-old mice. In livers, fluorescence intensity of DiR of 8-month-old mice was significantly higher than that of 8-week-old mice at 12 h. DiR accumulation in the heart, lung, spleen, and kidney was significantly slower than that in the liver and tumors within 24 h. DiR accumulation in the lung, kidney, heart, and spleen did not differ significantly between 8-month-old and 8-week-old mice. The volume of circulating blood in 8-month-old mice was 1.8-fold that of 8-week-old mice, with no significant difference observed in blood density (Table 3).

As mice age, the blood composition; the activity and quantity of other enzymes, such as P-450 enzymes in the liver; and the filtration capacity of the glomerulus are altered, thus affecting the distribution, metabolism, and excretion of administered formulations. With the increase in age, the immune capacity of 8-month-old mice was found to be stronger than that of 8-week-old mice. The number of PBMs in the peripheral blood of 8-month-old mice increased, the distribution of DE-SL (or DE-PL) in the liver and tumor changed, and the concentration of EPI in the peripheral blood did not change. For DE-SL, EPI accumulation in tumor was significantly increased by PBMs drug delivery system, and the concentration of EPI in liver was decreased. As the immune system was strengthened, the number of immune cells, including PBMs in the peripheral blood, were increased, and the prolonged circulation time of DE-PL decreased. The accumulation of DE-PL was significantly elevated in the liver and significantly reduced in the tumors of 8-month-old mice.

### 3.5. Pharmacodynamics of EPI-SL and EPI-PL in Aged and Young Mice

After the injection of EPI-SL and EPI-PL, the tumor growth rate of 8-month-old and 8-week-old mice slowed down significantly (Figure 6A). The hair color was dull and yellow, with weight loss observed after EPI-PL treatment (Figure 6B); 8-month-old mice had more PBMs in peripheral blood than 8-week-old mice, and the anti-tumor treatment effect of mice injected with EPI-SL was better than that of mice injected with EPI-PL (Figure 6C). During EPI-SL treatment, the tumor growth rate of treatment completion of 8M-SL was significantly slower than that of 8W-SL, and 8M-SL exhibited the best therapeutic effect. Fifty percent of 8M-SL mice exhibited tumor shedding. There was no sign of recurrence within 5 days of treatment completion. The tumor gradually scabbed and turned black. The black scab fell off after day 13, and mice hair returned by day 40, which was not significantly different from that observed in normal mice (Figure 6D). In addition, the survival rate of both 8-month-old and 8-week-old mice was maintained at 100%. Following EPI-PL treatment, the area of hair loss was significantly larger in 8-month-old mice than in 8-week-old mice, and the tumor growth rate of 8W-PL was significantly slower than that of 8M-PL (Figure 6E).

The number and uptake capacity of PBMs in 8-month-old mice were higher than those in 8-week-old mice. In the experiment of 8-week-old mice and 8-month-old mice, different drug delivery system (harnessing PBMs and evading immune system) produced different therapeutic effects [32]. With the increase of PBMs, the administration routes using PBMs drug delivery system increased the accumulation of EPI-SL in tumors [33]. For EPI-PL, the accumulation of EPI-PL at the tumor site is reduced. Due to the effect of long circulation, it accumulated in the hair follicles and liver, leading to hair loss and hepatitis, affecting the survival status of 8-month-old mice. Moreover, EPI-PL accumulated in the tumor by the EPR effect and could not improve the tumor microenvironment. After EPI-PL was stopped, the tumor growth rate accelerated [34,35].

### 3.6. Tissue Section Staining of EPI-SL and EPI-PL in Aged and Young Mice

Only small amounts of apoptotic cells were observed in the tumors of 8M-Con and 8W-Con; however, the density of apoptotic cells was significantly altered after EPI-PL and EPI-SL treatment. The tumor cell density of 8-month-old mice was lower after EPI-SL treatment than that in the other groups. Immune cells and liposomes could not easily access tumor cell growth areas of 8M-PL, 8W-SL, and 8W-PL (red circles, Figure 6F,G and 7A). After EPI-SL treatment, the heart, thymus, lungs, spleen, and kidneys of mice in all experimental groups showed no noticeable damage. After EPI-PL treatment, inflammatory cells were found to invade the heart and liver of 8-month-old and 8-week-old mice, and the red and white marrow in the spleen revealed blurred edges (red arrows). Compared with the treated 8-week-old mice, the middle cortex and medulla of the thymus of 8-month-old mice had a more explicit structure (Figure 7B).

The thymus, heart, liver, lung, spleen, and kidneys did not show any apparent damage. EPI-SL was carried and delivered to the tumor by monocytes, killing the tumor cells and inducing the tumor recruitment of additional monocytes in the tumor microenvironment [10]. SA better harnessed PBMs to deliver EPI-SL to 8-month-old immune-enhanced mice. The accumulated EPI in the hair follicles could not be eliminated in the period following the EPI-PL treatment, resulting in large areas of hair loss, similar to the hand-foot syndrome caused by Doxil^®^ [36]. EPI-PL passively accumulated in the tumor, which can be attributed to the prolonged circulation of EPI-PL and the EPR effect. When immunity increased, the prolonged circulation of EPI-PL decreased, and the tumor growth rate was significantly faster than that of treated 8-week-old mice. In 8M-PL, possible high-density growth areas of tumor cells could be present in the tumor; this could potentially cause tumor recurrence after treatment completion and increased accumulation in other tissues, resulting in inflammation and damage to the liver, spleen, and heart. The change in age decreased the efficacy of EPI-PL and increased the side effects of EPI-PL.

## 4. Conclusions

The experimental results were consistent with previous findings that changes in age would affect the anti-tumor pharmacodynamics of EPI-SL and EPI-PL. Due to an increase in age, the immune system was more mature, and 8-month-old mice had more PBMs in peripheral blood than 8-week-old mice. Furthermore, PBMs that expressed Siglec-1 receptors in the blood delivered more EPI-SL to the tumor, reducing tumor immunosuppression and enhancing immune cell infiltration into the tumor, thereby fundamentally treating the tumor and initiating tumor shedding. Compared with the treated 8-week-old mice, 8-month-old mice lost their hair after EPI-PL treatment, and the tumor inhibition rate decreased even with increased distribution of drugs in the liver and spleen. Overall, SA-modified liposomes inhibited tumor growth in 8-month-old mice by harnessing PBMs.

## Figures and Tables

**Figure 1 pharmaceutics-14-00545-f001:**
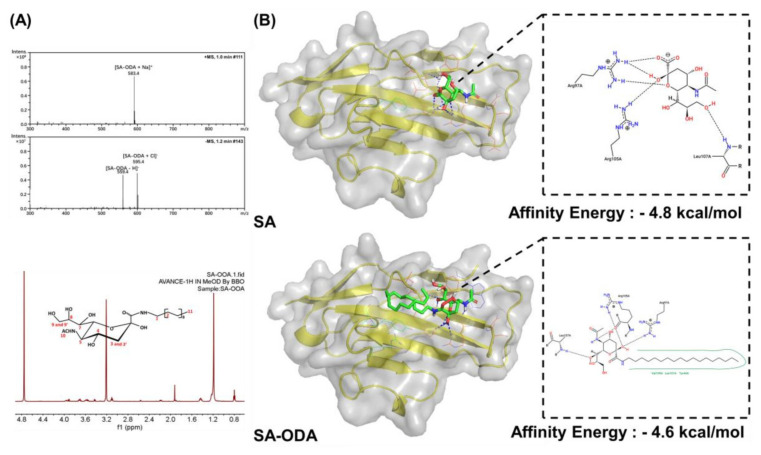
(**A**) Mass spectrum of the SA-ODA, and 1H NMR spectrum and chemical structure of the SA-ODA. SA-ODA, sialic acid-octadecylamine derivatives. (**B**) The specific recognition of Siglec-1 receptor and SA-ODA.

**Figure 2 pharmaceutics-14-00545-f002:**
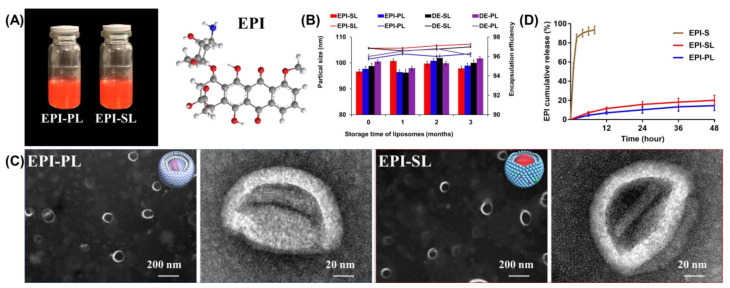
(**A**) The placement stability and appearance of the liposomes. Simulation of the interaction with SO_4_^2+^ and EPI. (**B**) The changes of EE% and particle size after three months, n = 6, (curve: particle size, histogram: EE%). (**C**) Phosphotungstic acid negative stain image of EPI-SL and EPI-PL. (**D**) EPI cumulation release of EPI-SL, EPI-PL, and EPI-solution, n = 3.

**Figure 3 pharmaceutics-14-00545-f003:**
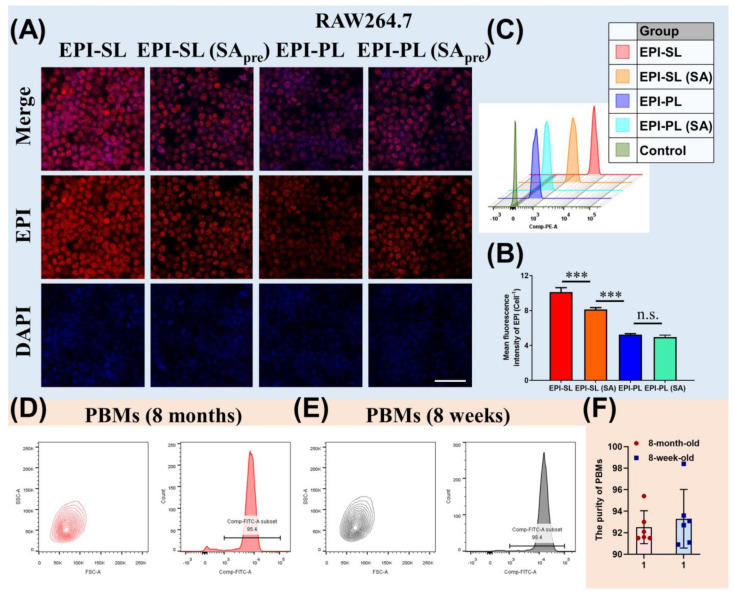
(**A**) When RAW264.7 cells and SA presaturated RAW264.7 cells were incubated with EPI-SL and EPI-PL for 2 h, the fluorescence picture of RAW264.7 was taken by a confocal microscope (red fluorescence: EPI. Scale bar = 100 μm), (**B**) and the fluorescence intensity of EPI in cells was measured by ImageJ (1 dm × 1 dm, n = 3, ^ns^
*p* ≥ 0.05, and *** *p* < 0.001). (**C**) When RAW264.7 cells and SA presaturated RAW264.7 cells were incubated with EPI-SL and EPI-PL for 2 h, total fluorescence picture of EPI in RAW264.7 cells was measured by a flow cytometer (10,000 cells). (**D**–**F**) The purity of PBMs extracted from aged and young mice (CD115-FITC, n = 6).

**Figure 4 pharmaceutics-14-00545-f004:**
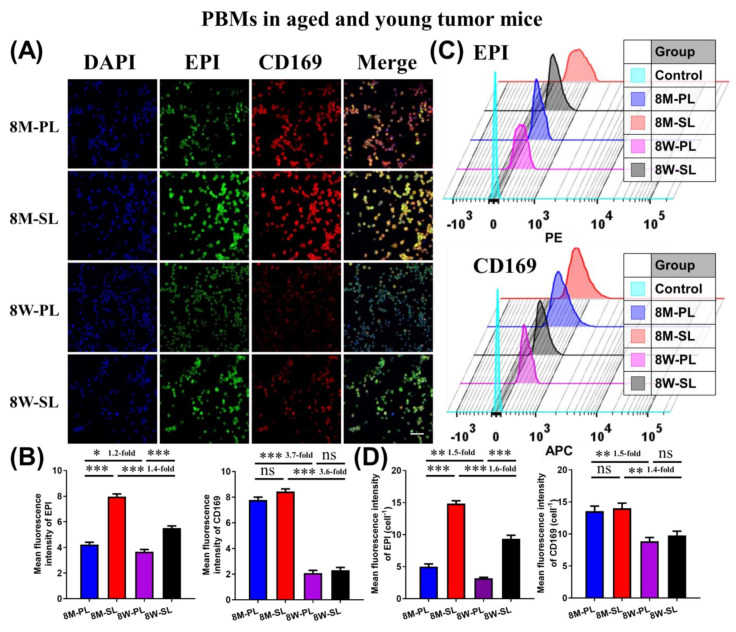
(**A**) When isolated Peripheral Blood Monocytes were incubated with EPI-SL and EPI-PL for 2 h, the fluorescence picture of isolated PBMs was taken by a confocal microscope (green fluorescence: EPI, red fluorescence: CD169. Scale bar = 100 μm), (**B**) and the fluorescence intensity of EPI and CD169 in isolated PBMs was measured by ImageJ (1 dm × 1 dm, n = 3, ^ns^
*p* ≥ 0.05, * *p* < 0.05, ** *p* < 0.01, and *** *p* < 0.001). (**C**) When isolated PBMs were incubated with EPI-SL and EPI-PL for 2 h, total fluorescence intensity of EPI and CD169 in PBMs was measured by a flow cytometer (10,000 cells). (**D**) Mean fluorescence intensity of EPI and CD169 in single cell.

**Figure 5 pharmaceutics-14-00545-f005:**
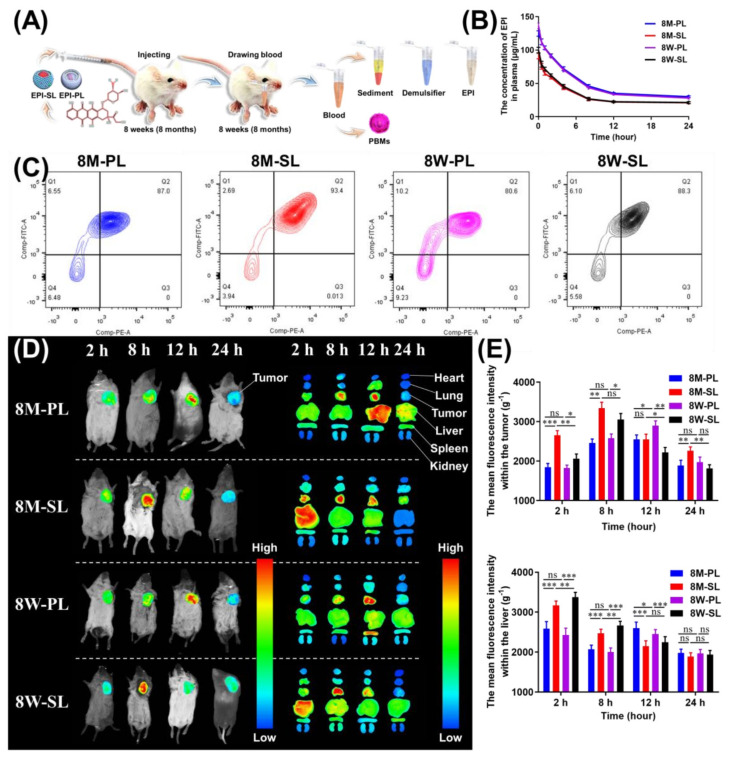
In vivo accumulation of EPI in organ of 8-month-old and 8-week-old S180 tumor-bearing mice. (**A**) The pharmacokinetic process in tumor-bearing mice at 8 weeks (or 8 months) for 24 h, volume of tumor is 100 cm^3^, n = 3. (**B**) The concentration of EPI in blood after treatment with EPI-PL and EPI-SL, n = 3. (**C**) Fluorescence intensity of EPI in PBMs after treatment with EPI-PL and EPI-SL for 2 h, FITC: CD115, PE: EPI, n = 3. (**D**) The fluorescence intensity of tumors in vivo and organs in vitro of different-aged mice. (**E**) The cumulative fluorescence intensity of DE-SL and DE-PL in the liver and tumor at 2, 8, 12, and 24 h (n = 3, ^ns^
*p* ≥ 0.05, * *p* < 0.05, ** *p* < 0.01, and *** *p* < 0.001).

**Figure 6 pharmaceutics-14-00545-f006:**
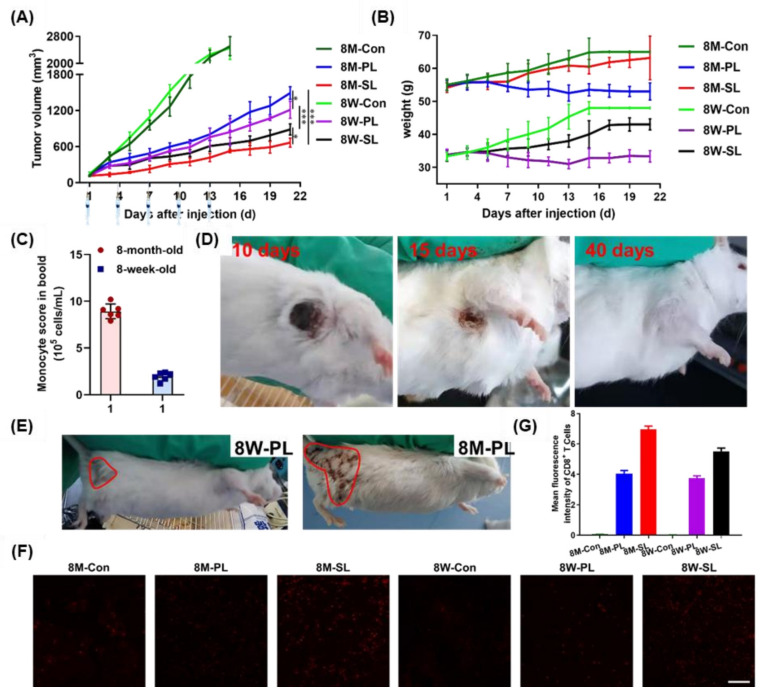
(**A**) Tumor volume of mice after completion of EPI-SL and EPI-PL treatment in tumor-bearing mice, n = 6, * *p* < 0.05, and *** *p* < 0.001. (**B**) Weight of mice after completion of EPI-SL and EPI-PL treatment in tumor-bearing mice, n = 6. (**C**) The number of PBMs in blood isolated from mice of different ages. (**D**) Tumor shedding of 8-month-old Kunming mice. (**E**) The area of hair shed between 8M-PL and 8W-PL. (**F**) The changes of CD8^+^ T cells in tumor in each treatment group, (**G**) and mean fluorescence intensity of CD8^+^ T cells (Scale bar = 100 μm).

**Figure 7 pharmaceutics-14-00545-f007:**
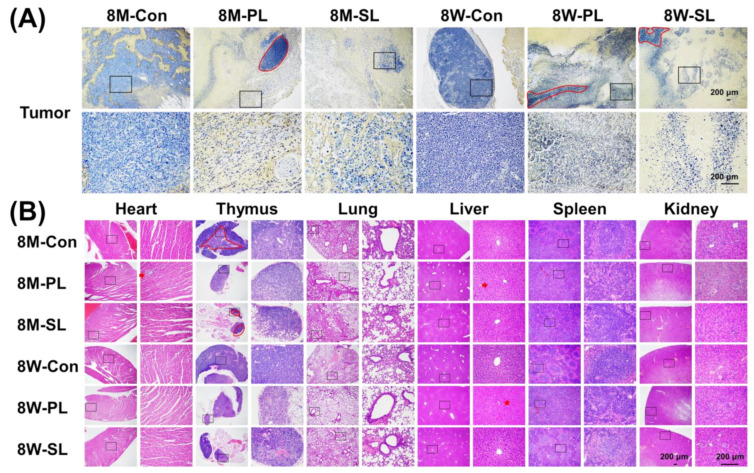
(**A**) The distribution of live and apoptotic cells in tumors of different-aged mice by using TUNEL (40×, 200×). (**B**) The degree of damage to heart, thymus, lung, liver, spleen, and kidney sections of different-aged mice using hematoxylin and eosin (H&E). The red marks indicate abnormal cell states (40×, 200×).

**Table 1 pharmaceutics-14-00545-t001:** Basic parameters of the liposomes (n = 3).

	Drug/Lipid (mg/mg)	Composition (n/n/n)	Particle Size (nm)	Zeta Potential (mV)	PDI	EE (%)
EPI-SL	1:10	SA-ODA/HSPC/CH = 5/50/45	110 ± 1	−18 ± 1	0.12 ± 0.01	96 ± 2
EPI-PL	1:10	PEG_2000_-DSPE/HSPC/CH = 5/50/45	108 ± 1	−31 ± 2	0.11 ± 0.02	96 ± 2
DE-SL	1:50	SA-ODA/HSPC/CH = 5/50/45	114 ± 1	−17 ± 1	0.11 ± 0.02	98 ± 3
DE-PL	1:50	PEG_2000_-DSPE/HSPC/CH = 5/50/45	109 ± 1	−31 ± 2	0.11 ± 0.01	98 ± 2

**Table 2 pharmaceutics-14-00545-t002:** The IC_50_ value of EPI-S, EPI-SL, and EPI-PL (n = 6).

	EPI-S (mg/L)	EPI-SL (mg/L)	EPI-PL (mg/L)
RAW 264.7	1.0 ± 0.1	4.2 ± 0.3	25 ± 1
S180	1.1 ± 0.1	7.5 ± 0.7	41 ± 3

**Table 3 pharmaceutics-14-00545-t003:** EPI pharmacokinetic parameters and blood parameters in 8M versus 8W mice (n = 3).

Kunming Mice	AUC_(0–24)_ (mg/L × h)	V_1_ (L/kg)	Weight of Mice	Volume of Peripheral Circulating Blood	Density of Peripheral Circulating Blood
8M-PL	1011 ± 28	0.057 ± 0.001	55 ± 1 g	1.89 ± 0.03 mL	1.03 ± 0.04 mg/mL
8M-SL	632 ± 23	0.059 ± 0.002	53 ± 1 g	1.86 ± 0.06 mL	1.05 ± 0.06 mg/mL
8W-PL	1092 ± 31	0.039 ± 0.001	30 ± 2 g	1.04 ± 0.04 mL	1.06 ± 0.05 mg/mL
8W-SL	639 ± 22	0.041 ± 0.001	33 ± 1 g	1.01 ± 0.02 mL	1.09 ± 0.06 mg/mL

## Data Availability

All data generated or analyzed during this study are included in this manuscript.

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
