# Peer review of "The Fate of Sialic Acid and PEG Modified Epirubicin Liposomes in Aged versus Young Cells and Tumor Mice Models"

_pharmaceutics, 2022, doi:10.3390/pharmaceutics14030545_

Round 1
Reviewer 1 Report
The manuscript " The fate of sialic acid and PEG modified Epirubicin liposomes in aged versus young cells and tumor mice models" can be accepted for publication in the Pharmaceutics after major revision. This research was interesting. Authors provided the new insights of new dimensional research approach through the development of novel nanoformulations. However, there was defective in the work. So, the manuscript under the current situation cannot be accepted for publication in Pharmaceutics. The authors need to address the following concerns.
- The catalog numbers of the reagents used are missing.
- The English used in the manuscript needs to be improved. The manuscript can be revised further for grammatical and typological errors.
- Formatting issues subscript, superscript, space between value and units need to be attended to.
- The abstract was not well drafted and can be rewritten.
- Did the authors perform the biocompatible or hemocompatibility studies of the formulation before proceeding to the in vivo assessment?
- FTIR analysis should be added in this research.
- Discussion is poorly. discussion needs to elaborated more (particularly in nano section).
- Did you assess the kinetic models of the materials for release study? should be done.
- Did you assess the stability of the formulations? should be done.
- The authors should further clarify the novelty of this study in both the Abstract and the last paragraph of the introduction. What the results of this study is adding to the field of cancer therapy, and why the other research studies didn't, and how these results can help others to take advantage in improving the design of their system. More comparison between this study and other previously published articles should be added (particularly in the Discussion section)
Author Response
Reviewer: 1
- The catalog numbers of the reagents used are missing.
Thank you for your nice suggestion. Due to the epidemic and the Spring Festival, I could not get the original files from the studio. Subsequent revisions would add catalog numbers of the reagents.
- The English used in the manuscript needs to be improved. The manuscript can be revised further for grammatical and typological errors.
Thank you for your nice suggestion. I had corrected.
- Formatting issues subscript, superscript, space between value and units need to be attended to.
Thank you for your nice suggestion. I had corrected.
- The abstract was not well drafted and can be rewritten.
Thank you for your nice suggestion. I had corrected.
- Did the authors perform the biocompatible or hemocompatibility studies of the formulation before proceeding to the in vivo assessment?
Thank you for your nice questions. I had performed anti-hemolysis and stability tests before proceeding to the in vivo assessment. After co-incubating SA and PEG-modified liposomes with blood for 48 h, the red blood cells did not undergo hemolysis, and the encapsulation efficiency of EPI did not change.
- FTIR analysis should be added in this research.
Thank you for your nice suggestion. The synthesis of SA-ODA was not the focus of this experiment. There had been previous articles in our laboratory explaining the synthesis of SA-ODA in detail. This study focused on the immune capacity change with aging, especially in function of peripheral blood monocytes for cancer therapy with liposomes modified with sialic acid or PEG.
- Discussion is poorly. discussion needs to elaborated more (particularly in nano section).
Thank you for your nice suggestion. I had added.
- Did you assess the kinetic models of the materials for release study? should be done.
Thank you for your nice questions. I did not assess the kinetic models of the materials for release study. There had been many release tests in the literature using this material.
- Did you assess the stability of the formulations? should be done.
Thank you for your nice questions. After SA and PEG-modified liposomes was placed at 4oC for 20 d, the encapsulation efficiency and particle size did not change.
- The authors should further clarify the novelty of this study in both the Abstract and the last paragraph of the introduction. What the results of this study is adding to the field of cancer therapy, and why the other research studies didn't, and how these results can help others to take advantage in improving the design of their system. More comparison between this study and other previously published articles should be added (particularly in the Discussion section)
Thank you for your nice suggestion. I had added. The experimental results were consistent with previous findings that changes in age would affect the anti-tumor pharmacodynamics of EPI-SL and EPI-PL. Due to an increase in age, the immune system was more mature, 8-month-old mice had more PBMs in peripheral blood than 8-week-old mice. Furthermore, PBMs that expressed Siglec-1 receptors in the blood delivered more EPI-SL to the tumor, reducing tumor immunosuppression and enhancing immune cell infiltration into the tumor, thereby fundamentally treating the tumor and initiating tumor shedding. Compared with the treated 8-week-old mice, 8-month-old mice lost their hair after EPI-PL treatment, and the tumor inhibition rate decreased even with increased distribution of drugs in the liver and spleen. Overall, SA-modified liposomes inhibited tumor growth in 8-month-old mice by harnessing PBMs. Significant differences in tumor molecular characteristics between patient populations of different ages can lead to a decrease in the anti-therapeutic effect of Nanoparticles.
Reviewer 2 Report
This study was performed to assess the influence of age on different anti-tumor mechanisms. For this purpose, anti-tumor effects of sialic acid-modified liposomes and poly(ethylene glycol) (PEG)-modified liposomes was compared in 8-week-old and 8-month-old mice. The results indicated that 8-month-old mice treated with sialic acid-modified liposomes achieved the best therapeutic effect. This age difference in therapeutic effect would be due to the drug deliver mediated by peripheral blood monocytes because 8-month-old mice had more peripheral blood monocytes in peripheral blood than 8-week-old mice. Review comments which should be addressed in the manuscript are followings,
- The abbreviation “PBMs” is defined as peripheral blood monocytes for the first time at line 104 on page 3 even though the abbreviation has been used many times above. The “PBMs” should be defined in the first appearance.
- It is not clear what are EPI-SL(SA) and EPI-PL (SA) in Figure 3 A. Does (SA) mean SA presented RAW264.7 cell? In addition, what means (Green fluorescence: EPI. Scale bar =100 µm) in the caption. I could not find green fluorescence and scale bar in the presented images in the Figure 3A. Authors should explain these points more and exactly.
- This study focus on the immune capacity change with aging, especially in function of peripheral blood monocytes for cancer therapy with liposomes modified with sialic acid or PEG. In fact, cancer is common in old people with weakened immune systems. It would be interesting to know what age range the 8-week-old and 8-month-old mice corresponds to in human. This point should be important to discuss how this study can be extrapolated for patients with cancer in different age groups.
Author Response
- The abbreviation “PBMs” is defined as peripheral blood monocytes for the first time at line 104 on page 3 even though the abbreviation has been used many times above. The “PBMs” should be defined in the first appearance.
Thank you for your nice suggestion. I had corrected, the “PBMs” should be defined in the first appearance.
- It is not clear what are EPI-SL(SA) and EPI-PL (SA) in Figure 3 A. Does (SA) mean SA presented RAW264.7 cell? In addition, what means (Green fluorescence: EPI. Scale bar =100 µm) in the caption. I could not find green fluorescence and scale bar in the presented images in the Figure 3A. Authors should explain these points more and exactly.
Thank you for your nice questions. I had corrected. (SA) mean SA presented RAW264.7 cell, red fluorescence: EPI. Scale bar =100 µm.
- This study focus on the immune capacity change with aging, especially in function of peripheral blood monocytes for cancer therapy with liposomes modified with sialic acid or PEG. In fact, cancer is common in old people with weakened immune systems. It would be interesting to know what age range the 8-week-old and 8-month-old mice corresponds to in human. This point should be important to discuss how this study can be extrapolated for patients with cancer in different age groups.
Thank you for your nice questions. This study focused on the immune capacity change with aging, especially in function of peripheral blood monocytes for cancer therapy with liposomes modified with sialic acid or PEG. Differences in antitumor therapy of SA and PEG-modified liposomes in different age groups. Rejuvenation of tumor patients is a general trend, and unified chemotherapy for different ages is the ultimate goal of this study. Through this experiment, it can be found that SA-modified liposome has a good therapeutic effect in all age groups.
Reviewer 3 Report
The paper “The fate of sialic acid and PEG modified epirubicin liposomes in aged versus young cells and tumor mice models “ by Sui et al. describes the preparation of two liposome system to test the effect of age on tumor growth. The paper is very well organized, several aspects of the research work have been accurately described. In my opinion, few minor changes need:
- Please revise sentence line 33 removing “And” at the beginning of the sentence
- I suggest to use the full the first time PBMs appears in introduction (line 42)
- Please revise sentence line 33 removing “And” at the beginning of the sentence
- Move Table 1 to Results and Discussion
- Please correct all errors in table 1, i.e.: 110 ± 1.22 should be written as 110 ± 1 or -17.8 ± 1.3 should be -18 ± 1 or 0.123 ± 0.012 should be 0.12 ± 0.01, etc
- Please revise sentences line 228-230, they seem to be copied by a protocol
- Please revise sentence line 234
- Please add ppm after H-NMR peaks
- Please reformulate sentence 290-291, what do you mean with “coupled the carboxyl group of hydroxyl group”?
- Please correct all errors in table 2, i.e.: 7.53 ± 0.69 should be 7.5 ± 0.7 or 62 ±3.02 should be 42± 3, etc
- Please correct all errors in table 3
Author Response
Reviewer: 3
- Please revise sentence line 33 removing “And” at the beginning of the sentence
Thank you for your nice suggestion. I had removed “And”.
- I suggest to use the full the first time PBMs appears in introduction (line 42)
Thank you for your nice suggestion. I had corrected.
- Please revise sentence line 33 removing “And” at the beginning of the sentence
Thank you for your nice suggestion. I had removed “And”.
- Move Table 1 to Results and Discussion
Thank you for your nice suggestion. I had moved Table 1 to Results and Discussion.
- Please correct all errors in table 1, i.e.: 110 ± 1.22 should be written as 110 ± 1 or -17.8 ± 1.3 should be -18 ± 1 or 0.123 ± 0.012 should be 0.12 ± 0.01, etc
Thank you for your nice suggestion. I had corrected.
- Please revise sentences line 228-230, they seem to be copied by a protocol
Thank you for your nice suggestion. I had deleted.
- Please revise sentence line 234
Thank you for your nice suggestion. I had deleted.
- Please add ppm after H-NMR peaks
Thank you for your nice suggestion. Due to the epidemic and the Spring Festival, I cannot get the original files from the studio. But there are records of peaks in the text. According to 1H NMR (CD3OD, dppm), 3.93 (2H, H-6, H-4), 3.71 (2H, H-9, H-5), 3.58 (2H, H-90, H-8), 3.21 (1H, H-7), 3.10 (2H, H-1), 2.19 (1H, H-30), 1.92 (3H, H-10), 1.43 (1H, H-3), 1.19 (30H, alkyl), 0.8 (3H, H-11) were identified (Figure 1 A).
- Please reformulate sentence 290-291, what do you mean with “coupled the carboxyl group of hydroxyl group”?
Thank you for your nice question. I had corrected. Zhou et al. coupled the carboxyl group of CH and the hydroxyl group of SA to form an SA derivative.
- Please correct all errors in table 2, i.e.: 7.53 ± 0.69 should be 7.5 ± 0.7 or 62 ±3.02 should be 42± 3, etc
Thank you for your nice suggestion. I had corrected.
- Please correct all errors in table 3
Thank you for your nice suggestion. I had corrected.
Round 2
Reviewer 1 Report
The main concerns requested were addressed and manuscript accept in present form